# Therapeutic Drug Monitoring in Perianal Fistulizing Crohn’s Disease

**DOI:** 10.3390/jcm11071813

**Published:** 2022-03-25

**Authors:** Mir Zulqarnain, Parakkal Deepak, Andres J. Yarur

**Affiliations:** 1Department of Medicine, Medical College of Wisconsin, Milwaukee, WI 53226, USA; mzulqarnain@mcw.edu; 2Division of Gastroenterology, Washington University in St. Louis School of Medicine, St. Louis, MO 63110, USA; deepak.parakkal@wustl.edu; 3Division of Gastroenterology, Medical College of Wisconsin, Milwaukee, WI 53226, USA

**Keywords:** Crohn’s disease, therapeutic drug monitoring, anti-tumor necrosis factor, perianal fistulas, infliximab, ustekinumab, vedolizumab, adalimumab

## Abstract

Perianal fistulas are a common complication of Crohn’s disease (CD) that has, historically, been challenging to manage. Despite the strong available evidence that anti-tumor necrosis factor (anti-TNF) agents are useful in the treatment of perianal fistulizing Crohn’s disease (PFCD), a significant number of these patients do not respond to therapy. The use of therapeutic drug monitoring (TDM) in patients with CD receiving biologic agents has evolved and is currently positioned as an important tool to optimize and guide biologic treatment. Considering the treatment of PFCD can represent a challenge; identifying novel tools to improve the efficacy of current treatments is an important unmet need. Given its emerging role in other phenotypes of Crohn’s disease, the use of TDM could also offer an opportunity to enhance the effectiveness of available therapies and improve outcomes in the subset of patients with PFCD receiving biologics. Overall, there is mounting evidence that higher anti-TNF drug levels are associated with better rates of “fistula healing”. However, studies have been limited by their use of subjective outcomes and observational designs. Ultimately, further interventional, randomized controlled trials looking into the relationship between drug exposure and fistula outcomes are needed.

## 1. Introduction

Crohn’s disease (CD) is an increasingly prevalent chronic inflammatory bowel disease (IBD) characterized by the development of inflammation in the gastrointestinal tract [1,2]. Among those patients with Crohn’s disease, some develop perianal fistulizing Crohn’s disease (PFCD). This is a debilitating phenotype that can be seen in up to a third of patients [3,4,5]. Its incidence increases with distal disease and its presence is associated with an overall worse prognosis [3,6]. It can lead to significant pain, perineal disfigurement, and fecal incontinence. Furthermore, patients with severe, refractory disease may also require proctectomy and permanent ostomy [7]. A multidisciplinary approach that includes combined medical and surgical therapies guided by radiologic and endoscopic diagnostics has shown to have a higher success rate in managing this phenotype than medical therapy alone [8]. Although there are multiple medical options available for the management of PFCD, most of them are limited in their overall efficacy.

Over the last decade, anti-tumor necrosis factor (anti-TNF) agents, particularly infliximab (IFX), have demonstrated their effectiveness in this subset of patients and have become first-line medical therapy in the treatment of PFCD. However, as with luminal Crohn’s disease, a significant fraction of patients do not respond to therapy. This has led into the investigation of pharmacokinetic mechanisms of non-response, such as low drug levels and anti-drug antibodies with fistula healing. Observational studies have revealed limited evidence that the use of TDM in patients with PFCD on biologics may potentially have a role in improving outcomes. In this narrative review, we sought to summarize the current evidence behind those biologic therapies utilized in PFCD while highlighting the emerging role of TDM in patients presenting with this phenotype.

## 2. Evidence behind the Current Biologic Therapies Utilized in the Management of Perianal Fistulizing Crohn’s Disease

Although there is growing evidence supporting the role of biologics in PFCD, there is a significant gap in knowledge regarding the positioning, optimization, and use of the biologics that are now available. Within these agents, IFX is one of the most recognized options due to the availability of randomized controlled trials supporting its efficacy in this patient population. The exact role of the newer generations of biologics, such as vedolizumab (VDZ) and ustekinumab (UST), remains less clear [9,10].

The first double-blind, placebo-controlled trial which studied anti-TNFs in PFCD was published in 1999 by Present et al. [11]. The study included 94 patients who were randomized to induction therapy with IFX dosed at 5 mg/kg, 10 mg/kg, or placebo at 0, 2, and 6 weeks. The primary endpoint of the trial was defined as a reduction of 50% or more in the number of draining fistulas at two or more consecutive study visits, which were required to be 21 days apart. The resolution of a draining fistula was defined as the lack of drainage upon gentle finger compression. The authors found that 68% of those patients receiving IFX at a dose of 5 mg/kg and 56% of those receiving IFX at a dose of 10 mg/kg achieved this primary endpoint. This rate of effectiveness was significantly higher than the 26% seen in the group that received placebo. Complete response, defined as absence of drainage on two consecutive visits, was seen in 55%, 38%, and 13% of patients treated with IFX 5 mg/kg, IFX 10 mg/kg, and placebo, respectively. The response rate in this initial trial did not appear dose related, with the 5 mg/kg group having a higher rate of response than the 10 mg/kg group.

A subsequent multicenter, double-blind, randomized, placebo-controlled trial performed by Sands et al. investigated the efficacy of IFX as a maintenance therapy for PFCD [12]. In this study, patients received an open label induction regimen of IFX 5 mg/kg at weeks 0, 2, and 6. Those patients that responded to treatment were then randomized to continue IFX 5 mg/kg or placebo every 8 weeks for 54 weeks. Response to treatment was defined as a reduction of at least 50% from baseline in the number of draining fistulas at weeks 10 and 14 after the induction regimen. The primary outcome was loss of response through week 54 of treatment defined as: recurrence of draining fistulas, need for additional or alternate therapy for worsening or persistent (luminal) disease, need for surgery, or self-discontinuation of the medication by the patient due to lack of efficacy. At the end of the follow-up, those patients that received IFX therapy had a significantly lower rate of loss of response when compared to those that received placebo (more than 40 weeks vs. 14 weeks, respectively [*p* < 0.001]).

What makes these trials unique is that they are among the few randomized controlled trials investigating a biologic agent for PCFD. However, a considerable limitation is the subjective nature of the primary endpoint of “fistula closure”, as it was based on the investigators’ physical evaluation of the patients and did include a more objective endpoint such as cross-sectional imaging. Additionally, the endpoint of “50% reduction in fistulas from baseline” was also left to the clinician’s assessment and different clinicians may have interpreted this endpoint differently. In an ideal scenario, a clinical trial should have a more objective, centrally read primary outcome. This has been recognized by investigators and a recent systematic review noted that radiologic outcomes are becoming more increasingly incorporated into the primary endpoints of trials investigating PFCD [13].

The efficacy of another anti-TNF, adalimumab (ADA), was shown in a sub-group analysis of the CHARM trial which demonstrated that at week 26 of treatment, complete fistula closure (defined as closure of all fistulas that were draining at screening visits) was achieved in 30% in the ADA-treated patients versus 13% of those receiving placebo [14]. This difference remained significant through week 56. Most notably, all the patients that achieved complete fistula closure at week 26 remained in remission at week 56. Another retrospective cohort study conducted in 15 tertiary centers in Spain showed that at 6 months of therapy, 66% of patients treated with ADA experienced an improvement in complex fistulas [15]. A particular strength of this study was the use of magnetic resonance imaging to assess PFCD activity. The authors did find a correlation between clinical and radiological disease activity (κ = 0.68). A more recent meta-analysis consisting of seven studies and 379 patients found that 36% (95% CI: 0.31–0.41) of patients receiving ADA had obtained complete fistula closure (defined as no draining fistulas on examination) at follow up periods ranging from 4 to 56 weeks [16].

Unlike IFX and ADA, the role of other biologics, such as certolizumab-pegol, VDZ, and UST, is less certain. Sub-group analysis of larger studies such as the GEMINI II trial and registry data show that a higher percentage of patients on VDZ experience fistula closure when compared to placebo [17,18,19,20,21,22,23]. This association is supported by the recent ENTERPRISE trial, a phase 4, randomized, double- blind, multicenter trial which evaluated the efficacy of VDZ in PFCD in patients with CD with 1-3 MRI-confirmed perianal fistulas [10]. One arm received VDZ, 300 mg IV at weeks 0, 2, 6, 14, and 22, while the other arm received the same regimen with an additional VDZ dose at week 10. At week 30, 53.6% of all subjects included in the study achieved the primary endpoint of having greater than a 50% decrease from baseline in the number of draining perianal fistulae (defined as no longer draining despite gentle finger compression). The arm randomized to receive an extra dose of VDZ at week 10 did not have better outcomes when compared to the arm that had the standard treatment regimen. However, despite the strength of being a prospective study, the generalizability of this trial was limited due to its small sample size and lack of a placebo-arm.

Regarding ustekinumab, the recent BioLAP multicenter retrospective study showed that, out of patients who had been received ustekinumab (UST) therapy for at least 3 months, 38.5% with active PFCD at initiation of treatment reached the endpoint of “clinical success” at 6 months [9]. Clinical success was defined by the absence of draining purulent material as determined by a clinician, as well as not having a need for new medical or surgical intervention. This study also showed that, among patients with a history of PFCD that was inactive at the time of ustekinumab initiation, only 22% had a recurrence of perianal disease. Despite the encouraging results, the study is limited by its poor definition of fistulas and retrospective design.

## 3. Therapeutic Drug Monitoring: Current Application in Inflammatory Bowel Disease

Despite the proven efficacy of biologics in IBD, up to 30% experience primary non-response, while approximately another 40% develop secondary non-response over time, requiring dose optimization or the need to switch therapy. The development of immunogenicity against the drug and/or sub-optimal drug levels can explain a significant number of therapeutic failures [24].

The advent of therapeutic drug monitoring, defined as the evaluation of serum drug concentrations and the presence/titers of anti-drug antibodies at a specific point in time has helped clinicians guide treatment by allowing them to identify those patients that may experience a benefit with dose optimization versus those where increase the dose is likely futile and should switch therapies and/or strategies. Numerous studies have demonstrated that higher serum biologic concentrations are associated with improved objective therapeutic outcomes, such as mucosal healing and normalization of inflammatory markers [25,26,27,28,29,30,31,32]. However, these studies have shown an association and not causation. Drug levels may be lower in patients with a higher disease burden and higher drug clearance. An important debate has been regarding the use of “pro-active” TDM of anti-TNFs, where drug doses are adjusted with the goal of maintaining a specific drug threshold, independently of disease activity [33,34]. The results have been conflicting, mainly due to the heterogenicity in the characteristics of patients, potential difference in target drug levels, and limitations in study design, among others.

## 4. Anti-Tumor Necrosis Factor Drug Levels and Outcomes in Perianal Fistulizing Crohn’s Disease

Studies looking into TDM and the association of anti-TNF drug levels and drug efficacy have also been conducted in sub-group of CD disease patients with specific phenotypes (Table 1). Emerging evidence has shown a strong association between higher anti-TNF drug levels and fistula healing in PFCD. A retrospective cohort study by Davidov et al. included 36 patients with active PFCD who received IFX at a standard dose of 5 mg/kg at weeks 0, 2, and 6, followed by every 8 weeks and looked at the association of drug levels and clinical response at week 14, defined as “decreased drainage of fistulas as reported by the patient and verified by a physician” [35]. The authors found that the group of patients with “clinical response” had higher median trough IFX levels when compared to those that did not (week 2, 20 vs. 5.6 µg/mL, *p* = 0.0001; week 6, 13.3 vs. 2.55 µg/mL *p* = 0.0001; and week 14, 4.1 vs. 0.14 µg/mL, *p* = 0.01). Specifically, IFX serum levels ≥9.25 µg/mL at Week 2 and ≥7.25 µg/mL at Week 6 were noted to be best associated with response to treatment at week 14. Despite its positive findings, this study had several limitations including a small sample size, retrospective study design, lack of follow up fistula imaging on most patients and the inherent subjective nature in which the outcomes were measured.

A larger retrospective study performed by Yarur et al. included 117 PFCD patients with an active fistula and showed that, at a median of 29 weeks of IFX therapy, those with healing of perianal fistula (defined as absence of drainage after gentle compression) had higher trough IFX concentrations in comparison to those with active disease (15.8 μg/mL versus 4.4 μg/mL; *p* < 0.001) [36]. Quartile analysis of serum IFX concentrations showed that IFX levels >10.1 μg/mL and >20.3 μg/mL were associated with three- and eight-fold chance of fistula healing, respectively. Additionally, patients with fistula healing had a lower likelihood of having serum anti-IFX antibodies (OR, 0.04; 95% CI, 0.004–0.3, *p* < 0.0001) and IFX levels ≥10.1 mcg/mL were significantly associated with fistula closure (OR, 2.9; 95% CI, 1.1–8.7, *p* < 0.036). Notably, a subset of included patients achieved fistula healing only at levels of ≥20 mcg/mL, which potentially supports the approach of optimizing drug levels to this threshold prior to abandoning therapy in patients who have not experienced fistula healing at lower trough IFX levels. The study was limited given its retrospective study design and due to its failure in distinguishing simple vs. complex fistulas. As in other TDM studies, the results only proved an association and not causation. A particular strength of the study was that it contained the largest sample size of any study investigating this topic and did include patients who had received dose optimization/escalation, opening the possibility to assess the rates of fistula healing on those patients with a high IFX exposure.

Strik et al. added to this growing body of literature with a retrospective study investigating ADA in addition to IFX in the treatment of PCFD [37]. Patients maintained on these anti-TNFs were separated into two groups based on the status of their fistulas (actively draining or non-draining). Fistula closure was defined as the absence of purulent discharge upon gentle finger compression and/or fistula closure on MRI of the pelvis. The authors found that serum trough levels were significantly higher in patients with fistula closure as compared to those with active drainage in both IFX (6.0 μg/mL vs. 2.3 μg/mL; *p* < 0.001) and ADA groups (7.4 μg/mL vs. 4.8 μg/mL; *p* = 0.003). An IFX trough level of ≥5 µg/mL and an ADA trough level ≥5.9 was significantly associated with perianal fistula closure. For IFX, higher closure rates were seen in those naïve to biologics and with combination therapy as opposed to the patients receiving monotherapy. In the ADA group, the treatment duration and combined use of a seton was associated with higher rates of fistula closure. The objectivity provided by MRI (as opposed to the subjectivity of physical exam) was a particular strength of this study.

Plevris et al. also investigated both IFX and ADA for PFCD with a retrospective cross-sectional study including 64 patients on maintenance therapy for at least 24 weeks [38]. Drug levels were measured ±4 weeks of the clinical assessment of the fistula. IFX drug and antibody levels were measured at trough, while for ADA drug and antibody levels were measured at any time between doses. The primary outcome was perianal fistula healing (defined as the absence of drainage) and the secondary outcome was perianal fistula closure (defined as no external skin opening in the peri-anal area). Patients with fistula healing had higher levels of anti-TNF trough levels vs. those without fistula healing (ADA: 12.6 vs. 2.7 μg/mL, *p* < 0.01; IFX: 8.1 vs. 3.2 μg/mL, *p* < 0.01). Patients with fistula closure also had significantly higher anti-TNF trough levels vs. those without fistula closure (ADA: 14.8 vs. 5.7 μg/mL, *p* < 0.01; IFX: 8.2 vs. 3.2 μg/mL, *p* < 0.01). Receiver operating characteristic analysis revealed a cutoff of ≥6.8 μg/mL for fistula healing and ≥9.8 μg/mL for fistula closure in patients receiving ADA and an optimum trough of ≥7.1 μg/mL for both fistula healing and closure for IFX. Again, the retrospective design of the study and lack of objective evaluation of the fistulas with imaging were limitations of this study.

El-Matary et al. performed a multicenter prospective cohort study including 27 pediatric patients (<17 years) with PCFD who were treated with IFX and who had serum trough drug titers measured before the fourth dose [39]. The median IFX pre-fourth dose level in the responders (defined as a decrease in drainage of fistulas) was 12.7 ug/mL, compared with 5.4 ug/mL in the group with no response (*p* = 0.02). A particular strength of this study was its prospective study design; however, the small sample size, lack of long term follow up, and the subjective primary outcome were notable limitations.

In a post-hoc analysis, Ruemmele et al. performed a sub-analysis of the data from the IMAgINE 1 and IMAgINE 2 trials [40]. These trials cumulatively followed pediatric patients for 292 weeks and demonstrated the efficacy of ADA in fistula closure and fistula improvement (as defined as closure of all baseline fistulas or decrease in number by ≥50%, respectively, for at least two consecutive visits). The patients were randomly assigned to receive either high dose ADA (defined as 20 mg every other week [EOW] or 40 mg EOW if >40 kg) or standard doses (defined as 10 mg every other week [EOW] or 20 mg EOW if >40 kg). Although the concentration of ADA in patients with fistula closure trended slightly higher than those not achieving fistula closure at weeks 16 (7.4 µg/mL vs. 7.0 µg/mL) and 52 (10 µg/mL vs. 6.1 µg/mL), there was no statistically significant difference between the two groups. While this contradicted the adult studies and findings of El Matary et al., the limited study was not powered to detect statistical differences between treatment groups, did not have a placebo-arm, and lacked objective assessment of fistula closure with pelvic MRI.

The most recent evidence supporting optimizing post-induction IFX levels arises from a post hoc analysis of the ACCENT-II trial by Papamichael et al., which evaluated patients with fistulizing CD receiving induction and maintenance infliximab therapy [41]. Measured outcomes included fistula response (defined as a reduction of at least 50% of draining fistulas from baseline), complete fistula remission (defined as absence of draining fistulas), CRP normalization (defined as a CRP level ≤5 mg/L), and, finally, a composite outcome of both complete fistula remission combined with CRP normalization at week 14 and week 54. Higher week 14 IFX concentrations were independently associated with week 14 fistula response (odds ratio [OR]: 1.16; 95% confidence interval [CI]: 1.02–1.32; *p* = 0.019), and composite remission (OR: 2.32; 95% CI: 1.55–3.49; *p* < 0.001). Higher week 14 IFX concentrations were also independently associated with week 54 composite remission (OR: 2.05; 95% CI: 1.10–3.82; *p* = 0.023). ROC curve analysis identified an IFX concentration of ≥9.6 μg/mL at week 6 to be associated with complete fistula response at week 54. Most notably, the analysis revealed that IFX concentrations of ≥26.1 μg/mL at week 6 and ≥8.7 μg/mL at week 14 were associated with the highest rates of early composite remission (36% and 48%, respectively). Furthermore, IFX concentrations ≥11.3 μg/mL at week 14 were associated with the highest rate of long-term composite remission. These findings are interesting and open the debate on whether proactively increasing infliximab doses early on therapy may improve short- and long-term outcomes. Randomized controlled trials are warranted to support this hypothesis.

A cross-sectional retrospective study by De Gregario et al. added a more objective viewpoint to existing evidence by documenting the association of anti-TNF levels and radiologic fistula outcomes [42]. This study included 193 patients with PFCD on maintenance IFX or ADA who had drug levels checked within 6 months of a pelvic MRI. Radiologic disease activity was scored using the Van Assche Index (VAI) with an inflammatory subscore calculated using multiple indices: T2-weighted imaging hyperintensity, collections >3 mm diameter, and rectal wall involvement. The primary endpoint was radiologic healing (inflammatory subscore ≤6). The secondary endpoint was radiologic remission (inflammatory subscore = 0). Patients with radiologic healing had higher median drug levels compared with those with active disease (IFX 6.0 vs. 3.9 μg/mL; ADA 9.1 vs. 6.2 μg/mL; *p* < 0.05 for both). Patients with radiologic remission also had higher median drug levels compared with those with active disease (IFX 7.4 vs. 3.9 μg/mL; *p* < 0.05; ADA 9.8 vs. 6.2 μg/mL; *p* = 0.07). This study is unique because most of the other retrospective trials had a largely subjective definition of fistula healing and lacked the objectivity provided by imaging studies. Despite this distinguishing attribute, the study does have multiple limitations. The VAI is not a validated scoring index and since the imaging was not centrally reviewed, there was inherent risk of variability and bias from the different radiologists interpreting the images.

Prospective randomized studies looking into the association of fistula healing and VDZ and UST serum levels are scarce. Data from the ENTERPRISE study did show that, in the group that received VDZ with an extra dose at week 10, patients with fistula healing had a higher pooled VDZ trough concentration between weeks 6 and 22 of the study. However, since more patients terminated treatment early in this group, and given the overall small sample size, the authors were unable to draw a definitive conclusion regarding the relationship between VDZ drug exposure and treatment response [10].

## 5. Discussion

The management of PFCD typically combines a medical and surgical approach. Biologics, especially anti-TNF agents, have demonstrated an important role in the treatment of PFCD. The current evidence supports an association between higher IFX and ADA serum drug levels with higher rates of fistula healing. Considering these findings, it would be tempting for many clinicians to assume that increasing drug doses could improve outcomes. However, randomized controlled trials are needed to prove this hypothesis. The lower drug levels seen in those patients that do not achieve fistula healing could potentially be explained by a higher inflammatory burden and higher drug clearance. The current literature is also significantly limited by deficits in study design, low sample sizes, variability in patient selection, failure to stratify different types of fistulas, and lack of objective endpoints.

One ongoing interventional randomized controlled study by Gu et al. may offer a better insight on how TDM could effectively be used in PFCD [43]. The PROACTIVE trial (Prospective randomized controlled trial of adults with perianal fistulizing Crohn’s disease and optimized therapeutic IFX levels) is enrolling patients with active PFCD randomized to either a proactive TDM group or standard dosing group with a 54 week follow up period. The proactive TDM group will have IFX dosing optimized to target higher trough concentrations at various time points (≥25 µg/mL at week 2, ≥20 µg/mL at week 6 and ≥10 µg/mL during maintenance therapy). The standard arm will be treated with the standard 5 mg/Kg dose of IFX at weeks 0, 2, and 6 weeks followed by every 8 weeks. The primary outcome of the study will be fistula healing at week 32 and secondary outcomes include fistula closure, fistula healing, radiological fistula healing, economic costs, and patient-reported outcomes. The addition of a radiologic outcome will serve to support the more subjective, clinical primary outcome. This is helpful as many of the studies reviewed lack this level of objectivity.

This randomized trial may also help prove causation and not just correlation when it comes to increased drug levels and improved fistula healing. Currently there is a strong association between the two, but the decreased serum levels in non-healing fistulas may be due to other factors, such as increased drug clearance and a higher inflammatory burden. This concern is supported by the ATLAS study which showed that there are also localized tissue factors that play a role in variations of local and systemic drug levels [44]. This study was unique in that it not only reported an accurate measurement of tissue levels of anti-TNF drug in luminal Crohn’s disease, but also found that these levels correlated well with the serum drug levels. The key finding of the study suggested that areas of severe luminal inflammation act as a ‘sink’ for the drug and resulted in diminished localized tissue drug levels. This drop in specific tissue drug levels may, thereby, be reducing its concentration and, therefore, efficacy in another area of inflammation, thus, leading to a mismatch in serum and tissue drug levels in these patients. The authors proposed this as an explanation for why patients with “normal” serum drug levels may still have uncontrolled disease and suggested that increasing these levels may result in improved outcomes. More recently, the same authors observed that increased serum infliximab levels were also associated with improved fistula healing in PFCD as compared to those with luminal disease. This leads to the question of possibly insufficient drug concentrations within the perianal tissues as a possible mechanistic explanation in treatment failure.

This question was recently explored by a pilot study assessing the fistula tissue levels of anti-TNF agents (infliximab and adalimumab) by use of ultraperformance liquid chromatography-mass spectrometry [45]. The authors obtained tissue samples from the fistula tracts of seven patients with Crohn’s perianal disease (five patients on adalimumab and two patients on infliximab) on maintenance treatment and compared tissue drug activity to negatives controls and spiked positive controls. They observed a lack of drug activity in all fistula samples taken from Crohn’s on maintenance therapy despite activity in positive controls. This raises the question on how tissue and serum pharmacokinetics are related and what role that the administration of higher doses may have on outcomes.

Higher dosing sub-groups do not always have better outcomes, as seen in the study cited above by Present et al. [11]. Although increasing dose and shortening intervals between doses has been shown to increase drug levels and clinical response in luminal disease there is significant pharmacologic variation between individuals. Patient characteristics such as high body weight, low albumin, and presence of ATI have been documented factors in increasing clearance of serum anti-TNF and leading to decreased serum levels [46]. There is also evidence that suggests shortening dose intervals may be a better way in increasing serum drug concentrations as opposed to simply increasing the medication dosage, especially in patients with low serum albumin levels. The development of an optimal, individualized dosing strategy for PFCD must consider all of these factors.

Another major question that warrants further investigation is the role that non-anti-TNF biologics can play in the treatment algorithm of PFCD and how the use of TDM for those drugs may help to optimize therapy in this patient population. Aside from observational evidence showing a possible dose-related response, there are limited randomized controlled data to guide the incorporation of these agents into the management of PFCD.

A recent multicenter randomized, controlled trial (ENTERPRISE) supported the use of VDZ in the treatment algorithm of PFCD by demonstrating a 53.6% pooled success rate in achieving the primary endpoint of having greater than a 50% decrease from baseline in the number of draining perianal fistulae [10]. This study also boasted a 71.4% fistula closure rate during the 30 week follow up period. Additionally, patients that responded to treatment trended towards a higher VDZ trough concentration between weeks 6 and 22 in the study arm that administered an extra dose of VDZ at week 10. However, despite these significant findings, the study was substantially limited by its lack of placebo-arm, small sample size (*n* = 38) and inability to support the findings with radiographic data.

No randomized controlled trials have assessed the efficacy of UST in PFCD; however, the recently published BioLAP multicenter, retrospective study showed that, out of patients who had been received UST therapy for at least 3 months, 38.5% with active PFCD at initiation of treatment reached the endpoint of “clinical success” at 6 months [9]. The interpretation of these data is limited by the subjective definition of the main outcome as it relied on a physician’s interpretation of fistula drainage and not a more objective outcome such as radiographic healing. Additionally, due to the retrospective nature of this study, the authors were not able to assess the relationship between serum UST levels and fistula response. They did find, however, that the lack of optimization of UST was associated with improved outcomes, but this was attributed to the refractory nature of disease in those that required aggressive drug optimization. Given this potential confounder and lack of drug level comparison between responders and non-responders in this study, the role of TDM with UST and PFCD remains unclear. Although this study shows a definite correlation between UST use and fistula healing, the exact role of UST in the treatment of PFCD remains uncertain and further prospective, randomized studies are needed.

Lastly, although this review focuses on the optimization of medical therapy with TDM, it is important to remember the crucial role of surgery in the management of PFCD. A combined medical and surgical approach in managing PFCD has shown to have better outcomes than medical therapy alone [47,48,49]. Irrespective of the prescribed medical therapies, individualized interventions, such as abscess drainage, seton placement, fistulectomy, fistulotomy, ligation, and advancement flaps, may be needed and, therefore, surgical consultation should be obtained to further guide these decisions.

## 6. Conclusions

PFCD is a challenging and debilitating phenotype of CD that has been historically difficult to manage. Anti-TNF agents, especially IFX, have emerged as the cornerstone of medical management in these patients. High quality evidence supporting the efficacy of most biologics and the potential role of TDM in PFCD is limited. Overall, the evidence supports that higher anti-TNF drug levels correlate with higher efficacy; however, no high quality, interventional data are available. This is partly because performing high quality clinical trials in PFCD can be challenging and costly. Moreover, conducting, and interpreting TDM studies impose their own challenges. Drug level concentrations may vary between laboratories and assays, which limits the extrapolation and comparison of results. Moreover, endpoints may vary across studies and patient demographics and selection may also complicate the interpretation of the data. Despite these challenges, further investigations in TDM are undergoing and may lead to a future of individualized and optimized management in patients not only with PFCD, but with IBD in general.

## Figures and Tables

**Table 1 jcm-11-01813-t001:** Studies demonstrating association between increased biologic drug levels with fistula healing in PFCD.

Author (Year)	Population	No. of Subjects	Anti-TNF	Primary Outcome	Drug Concentration in Active Fistulas (μg/mL)	Drug Concentration in Healed/Closed Fistulas (μg/mL)	Strengths	Limitations
Davidov et al. [35] (2016)	Adults	36	IFX	Decrease in drainage of fistulas	Week 2: 5.6 µg/mL Week 6: 2.55 µg/mL Week 14: 0.14 µg/mL	Week 2: 20.0 µg/mL Week 6: 13.3 µg/mL Week 14: 4.1 µg/mL	Similar demographics in both groups	small sample size, no imaging, subjective outcome
Yarur et al. [36] (2016)	Adults	117	IFX	absence of drainage	4.4 μg/mL	15.8 μg/mL	Large sample size	Retrospective, didn’t distinguish simple vs. complex fistulas
Strik et al. [37] (2019)	Adults	47 IFX 19 ADA	IFX ADA	absence of discharge upon gentle finger and/or fistula closure on MRI	IFX: 2.3 μg/mL ADA: 4.8 μg/mL	IFX: 6.0 μg/mL ADA:7.4 μg/mL	Assessment with imaging	Retrospective, didn’t distinguish simple vs. complex fistulas
Plevris et al. [38] (2020)	Adults	29 IFX 35 ADA	IFX ADA	Absence of drainage	IFX:3.2 μg/mL ADA: 2.7 μg/mL	IFX: 8.1 μg/mL ADA: 12.6 μg/mL	Secondary outcome of fistula closure	Retrospective, no imaging
Et Matary et al. [39] (2019)	Pediatric	27	IFX	Decrease in drainage of fistulas	5.4 ug/mL	12.7 ug/mL	Prospective study	Small sample size
Ruemmele et al. [40] (2018)	Pediatric	36	ADA	Closure of baseline fistulas or decrease in number by ≥50%	Week 16: 7.0 ug/mL Week 52: 6.1 ug/mL	Week 16: 7.4 ug/mL Week 52: 10.0 ug/mL	Well defined endpoints	Not powered to detect statistical difference, not randomized, not placebo controlled
Papamichael et al. [41] (2021)	Adults	Induction group *n* = 282 maintenance group *n* = 139	IFX	Fistula response: reduction of at least 50% of draining fistulas from baseline	No Response: 4.0 μg/mL	Response: 5.7 μg/mL	large sample size, the use of stringent endpoints	No imaging assessment of fistula, not randomized
De Gregario et al. [42] (2021)	Adults	117 IFX 76 ADA	IFX	Radiologic healing (inflammatory subscore ≤6 on Van Assche Index)	IFX: 3.9 μg/mL ADA: 6.2 μg/mL	IFX: 6.0 μg/mL ADA: 9.1 μg/mL	Use of radiographic parameters	Not placebo controlled, not randomized
Schwartz, D. A et al. [10] (2021)	Adults	VDZ (16) VDZ +10 (18)	VDZ	≥50% decrease from baseline in the number of draining perianal fistulae at week 30	~33 μg/mL (pooled trough conc. week 10)	~28 μg/mL (pooled trough conc. week 10)	Multicenter- RCT, use of MRI	Small sample size, no placebo arm

## Data Availability

Not applicable.

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
