# Peer review of "Therapeutic Drug Monitoring in Perianal Fistulizing Crohn’s Disease"

_jcm, 2022, doi:10.3390/jcm11071813_

Round 1

Reviewer 1 Report

Thanks for the privilege to review this interesting review article regarding perianal fistulizing CD and TDM. There are other narrative reviews over this topic published in the literature. Interestingly, the senior author of this review is first author of one of the first articles over the topic, and this fact per se puts this review under a good position scientifically. Please find below some considerations on the manuscript.

  1. Abstract is concise and objective. Puts the full perspective of the manuscript into the eyes of the reader. No specific comments.

2. in topic 2, at the end of first paragraph, authors culd cite rerferences regarding studies with vedo and ustekinumab. This would be the enterprise study and the french study with UST (line 56 page 2), which are used as references in other parts of the manuscript.

Ustekinumab for Perianal Crohn's Disease: The BioLAP Multicenter Study From the GETAID. Chapuis-Biron C, Kirchgesner J, Pariente B, Bouhnik Y, Amiot A, Viennot S, Serrero M, Fumery M, Allez M, Siproudhis L, Buisson A, Pineton de Chambrun G, Abitbol V, Nancey S, Caillo L, Plastaras L, Savoye G, Chanteloup E, Simon M, Dib N, Rajca S, Amil M, Parmentier AL, Peyrin-Biroulet L, Vuitton L; GETAID BioLAP Study Group.Am J Gastroenterol. 2020 Nov;115(11):1812-1820. doi: 10.14309/ajg.0000000000000810.PMID: 33156100

Efficacy and Safety of 2 Vedolizumab Intravenous Regimens for Perianal Fistulizing Crohn's Disease: ENTERPRISE Study. Schwartz DA, Peyrin-Biroulet L, Lasch K, Adsul S, Danese S.Clin Gastroenterol Hepatol. 2021 Sep 29:S1542-3565(21)01042-9. doi: 10.1016/j.cgh.2021.09.028. Online ahead of print.PMID: 34597729 

3. Page 3 line 105, could include at the end of the paragraph of data with ADA the meta analysis published:

A Meta-Analysis of Adalimumab for Fistula in Crohn's Disease.

Fu YM, Chen M, Liao AJ.Gastroenterol Res Pract. 2017;2017:1745692. doi: 10.1155/2017/1745692. Epub 2017 Oct 24.PMID: 29204155    4. Table 1 in the study by Schwartz, abbreviation of VDZ is wrong in the 4th column (not VDX but VDZ)   5. Las paragraph of page 8, when association vs causation is discussed, authors could include speculation on variation of higher doses versus higher concentrations in serum. E.g. present study, where 10 mg/kg did not lead to better outcomes. Higher Level is different than higher doses. Individual variation and other factors could be speculated (albumin, tissue penetration,  etc).   6. Another important point in discussion would be to add a paragraph on influence of adequate and individualized surgical management of patients independently of the serum concentrations.    

Author Response

1. Did not change abstract as

2. moved up references in topic 2 and changed the order of the citations as suggested

3. Page 3 line 105, added the suggested reference.

4. Corrected VDX to VDZ

 5. Added a paragraph regarding serum levels and drug dosing as suggested

6. Added the paragraph regarding surgical management of PFCD despite medical therapies, however we  kept it brief as we thought this was outside the scope of the review.

Please see attached corrected draft

Reviewer 2 Report

This is an interesting review of the literature regarding Therapeutic Drug Monitoring in Perianal Fistulizing Crohn's Disease.
This paper analyzes a lot of papers and show a general point of view about the therapy.
It's well written and good argument.

Author Response

please see corrected draft based on the other reviewer. Have added a few references and expanded discussion.
